# Copper Ion Mediates Yeast-to-Hypha Transition in *Yarrowia lipolytica*

**DOI:** 10.3390/jof9020249

**Published:** 2023-02-13

**Authors:** Mengqu Ran, Guowei Zhao, Liangcheng Jiao, Zhaorui Gu, Kaixin Yang, Lishuang Wang, Xinghong Cao, Li Xu, Jinyong Yan, Yunjun Yan, Shangxian Xie, Min Yang

**Affiliations:** Key Laboratory of Molecular Biophysics of the Ministry of Education, College of Life Science and Technology, Huazhong University of Science and Technology, Wuhan 430074, China

**Keywords:** *Yarrowia lipolytica*, dimorphic transition, copper, transcriptional regulation

## Abstract

Copper is an essential element that maintains yeast physiological function at low concentrations, but is toxic in excess. This study reported that Cu(II) significantly promoted the yeast-to-hypha transition of *Yarrowia lipolytica* in dose-dependent manner. Strikingly, the intracellular Cu(II) accumulation was drastically reduced upon hyphae formation. Moreover, we investigated the effect of Cu(II) on the physiological function of *Y. lipolytica* during the dimorphic transition and found that cellular viability and thermomyces lanuginosus lipase (TLL) were both influenced by the Cu(II)-induced yeast-to-hypha transition. Overall, hyphal cells survived better than yeast-form cells with copper ions. Furthermore, transcriptional analysis of the Cu(II)-induced *Y. lipolytica* before and after hyphae formation revealed a transition state between them. The results showed multiple differentially expressed genes (DEGs) were turned over between the yeast-to-transition and the transition-to-hyphae processes. Furthermore, gene set enrichment analysis (GSEA) identified that multiple KEGG pathways, including signaling, ion transport, carbon and lipid metabolism, ribosomal, and other biological processes, were highly involved in the dimorphic transition. Importantly, overexpression screening of more than thirty DEGs further found four novel genes, which are encoded by YALI1_B07500g, YALI1_C12900g, YALI1_E04033g, and YALI1_F29317g, were essential regulators in Cu-induced dimorphic transition. Overexpression of each of them will turn on the yeast-to-hypha transition without Cu(II) induction. Taken together, these results provide new insight to explore further the regulatory mechanism of dimorphic transition in *Y. lipolytica*.

## 1. Introduction

Metal elements are essential for all living organisms and play an important role in the growth, metabolism, and product synthesis, including the composition of transporters, participation in electron transfer, synthesis of essential hormones and vitamins, and regulation of nucleic acids and protein activities [1,2,3]. Nonetheless, high concentrations of metal ions are toxic, or even lethal, to cells [4]. To maintain suitable cellular metal homeostasis, cells have developed detoxification mechanisms to resist metal virulence, which are (i) the cell wall structure absorbs metals and restricts their entry into the cell; (ii) metal transporters strictly control the transport of metal ions; (iii) cellular proteins chelate metal ions; (iv) intracellular metal ions are transported into vacuoles [5,6,7,8,9]. Furthermore, morphological transitions can be induced by metal ions and this can specifically be found in many different phyla of fungi [10,11], whereas the regulatory mechanisms of the transitions are still lacking in many fungal species.

Dimorphic transition, also known as dimorphism, is defined as the morphology of some fungi switches between the oval-shaped yeast form and pseudohyphae or hyphae [12]. Dimorphic transition involves many biological processes, including metabolic regulation, genome synthesis, cell division, signal transduction, etc. [13,14,15], and it, therefore, stands for a systematic change in the metal ion adaption mechanism of fungi. Generally, the dimorphic transition is beneficial to many fungi to forage for nutrients and to infect humans, plants, or other hosts [16,17]. Therefore, a better investigation of the effects of metals on fungal dimorphism will deepen our understanding of both metal homeostasis and pathogenic mechanisms of fungal infections.

*Yarrowia lipolytica*, an oleaginous yeast, is a superior host for the sustainable production of biofuels and industrial chemicals due to its ability to accumulate high amounts of lipids [18]. In addition, it has been explored for the high-level production of various non-lipid products [19]. Therefore, *Y. lipolytica* has many promising applications in agriculture, industry, and bioremediation [20,21,22,23]. *Y. lipolytica* is also a kind of dimorphic fungus. Some stress conditions, such as neutral–alkaline pH, high temperature, and hypoxia, can switch it from yeast to hyphae form, and the accumulation of β-carotene also induces a morphological transition in *Y. lipolytica* [24,25]. Given the dimorphism of *Y. lipolytica* is beneficial to respond to environmental stress, but brings difficulties to the development and application of *Y. lipolytica* [26,27], there is an urgent need to explore the regulatory mechanism of the morphological transition in *Y. lipolytica*.

Copper is a trace life element that functions as a cofactor of a variety of enzymes and drives a series of biochemical reactions in the cell, playing an important role in cell growth, metabolism, energy production, and signal transduction [28,29,30,31]. However, high concentration Cu(II) induces cell death by targeting lipoylated TCA cycle proteins [32]. *Y. lipolytica* is an inherently Cu(II)-resistant yeast and has been used as a bioremediation agent in adsorbing Cu(II) pollutants [33]. In this context, the Cu(II) resistance mechanism of *Y. lipolytica* has been previously investigated. For instance, it is found that insertion of the transcription factor CRF1 increases Cu(II) resistance in *Y. lipolytica* [34] and the change of melanin and phosphatases levels is considered as the strategy to tolerant Cu(II) toxicity [35,36]. However, how Cu(II) affects the dimorphic transition of *Y. lipolytica* is still unclear.

In this study, the effects on the dimorphic transition of various metal ions were investigated by measuring the growth curve, cell viability, and target product rate, and the result showed, for the first time, that Cu(II) significantly promoted the yeast to hyphae transition in *Y. lipolytica*. The total and intracellular Cu(II) accumulation of *Y. lipolytica* was also determined before and after hyphae formation. In addition, the RNA-seq data on Cu(II)-induced dimorphic transition in *Y. lipolytica* lay the foundation for uncovering the regulatory mechanism of dimorphic transition in *Y. lipolytica*.

## 2. Results

### 2.1. Cu(II)-Induced Yeast-to-Hypha Transition in Y. lipolytica in the Stable Growth Period

To find out the metal inducer to the dimorphic transition of *Y. lipolytica*, the cells were cultivated in the presence of 2 mM metal ions for 9 days. Microscopic analyses revealed that the yeast-to-hypha transition was only found in the culture with 2 mM CuSO_4_. By contrast, neither the cells growing without metal ions nor those growing with 2 mM MgSO_4_, CoCl_2_, MnSO_4_, ZnCl_2_, NiCl_2_, and FeCl_3_ (Figure 1A) change cell morphologies. The results indicated that Cu(II) can specifically induce yeast-to-hypha transition in *Y. lipolytica*.

Furthermore, we found that the capacity of Cu(II) in inducing dimorphic transition is sufficient in shorter cultivation time and fewer concentrations. While cells growing in the control culture (0 mM CuSO_4_) or cultivated with less amount of Cu(II) (0.5 mM, and 1 mM CuSO_4_) remained in yeast forms, hyphal cells, whose lengths were over 20 μm, were found in the cultures supplemented with 2 mM CuSO_4_ on the 5th day, accounting for 18% of all kinds of different type of cells (Figure 1B, first row). Hyphae formation occurred on the 6th day in the presence of 1 mM CuSO_4_ and on the 7th day in the presence of 0.5 mM CuSO_4_, with hyphal cells exhibiting 20% and 15% of all the cells, respectively (Figure 1B, second and third row). Moreover, the proportion of hyphal cells increased with time in the presence of 0.5, 1, and 2 mM CuSO_4_ (Figure 1B). The results not only confirmed the capacity of Cu(II) in inducing dimorphic transition of *Y. lipolytica*, but also suggested that the effect of Cu(II) is dose-dependent. Therefore, we subsequently investigated the Cu(II) effect with a broader range of concentration in 12-day cultivation. As shown in Table 1, the minimum effective concentration of CuSO_4_ in inducing dimorphic transition was approximately 0.003 mM, and the maximum effective concentration was approximately 6 mM. In the range of effective concentrations, the transition can be accelerated by increasing Cu(II) concentration or achieved by prolonging culture times. While in concentrations of CuSO_4_ higher than 6 mM, the yeast-to-hypha transition is not found throughout the cultivation (Table 1). The results indicate that the dose-dependent effect of cooper ion on dimorphic transition exists only in such a concentration range.

Taken together, these findings indicate that Cu(II) specifically promotes the yeast-to-hypha transition of *Y. lipolytica*.

### 2.2. Cu(II) Accumulation of Y. lipolytica Cells Decreased after Hyphae Formation

To trace the fate and the accumulation of Cu(II), we measured the total (surface attached and intracellular) and intracellular Cu(II) concentrations in a series of cultures during the yeast-to-hypha transition. As shown in Figure 2A, the dry cell weight (DCW) of *Y. lipolytica* was drastically increased on the 5th day in the presence of 2 mM CuSO_4_, and on the 6th day in the presence of 1 mM CuSO_4_ and on the 7th day in the presence of 0.5 mM CuSO_4_. All time points correspond to the onset of showing dimorphic transition (see also, Figure 1B). By contrast, DCW of the control, which remained in yeast form, increased slowly (Figure 2A). The results suggested that Cu(II)-induced dimorphic transition is beneficial to biomass accumulation of *Y. lipolytica*. However, the total Cu(II) accumulations of *Y. lipolytica* cells were drastically decreased as a function of dimorphic transition (Figure 2B). Strikingly, the intracellular Cu(II) accumulation of *Y. lipolytica* cells was evidently increased before hyphae formation, while decreased after hyphae formation (Figure 2C). It seems that the elevated intracellular Cu(II) accumulation is a signal to induce dimorphic transition. Furthermore, the total and intracellular Cu(II) accumulation drastically decreased after hyphae formation due to increased biomass, which may be a strategy in responding to Cu(II) in *Y. lipolytica* (see Discussion). 

### 2.3. Cu(II) Reduces the Viability and the Thermomyces Lanuginosus Lipase (TLL) Activity of Y. lipolytica

Subsequently, we investigated the effect of Cu(II) on the physiology of *Y. lipolytica* during the yeast-to-hypha transition. Firstly, the overall cell viability was analyzed by CFU enumeration. The results showed that the overall viability of the cells grown in the medium containing 0.5, 1, and 2 mM CuSO_4_ were all significantly lower than the control cells without Cu(II) supplementation during days 4–8 (Figure 3A).

Secondly, trypan blue staining assay was employed to distinguish the proportion of membrane-impaired yeast-form cells, and the results showed that the proportion of viable yeast-form cells was decreased in the presence of Cu(II), as compared to the absence of Cu(II) (Figure 3B) during days 4~8. Notably, most hyphal cells in Cu(II) treatment was not penetrated by trypan blue, suggesting that the yeast-to-hypha transition is beneficial in maintaining cell membrane completeness (Figure 3B, inset).

Since thermomyces lanuginosus lipase (TLL) is an important lipase that has been widely expressed in *Y. lipolytica* [37,38], we subsequently examined the effect of Cu(II) on TLL activity. After quantifying the TLL activity during the dimorphic transition, we found that the TLL activity of *Y. lipolytica* decreased in the presence of 0.5~2 mM CuSO_4_ for 4 to 8 days, as compared with the control, although the TLL activities of all treatments increased day by day (Figure 3C). Notably, the cell-free enzymatic analysis showed that the TLL activity did not significantly change by the presence of 0.5~2 mM CuSO_4_ in vitro (Figure 3D), suggesting that Cu(II) itself did not affect cellular TLL activity directly. 

Taken together, these results indicate that Cu(II) had a toxic effect on the cells, which resulted in the reduction of the viability and lipase activity.

### 2.4. Overall Gene Expression Changes in Cu-Induced Yeast-to-Hypha Transition

To further investigate how Cu(II) promotes the dimorphic transition, we performed transcriptomic analysis (RNA-Seq) of *Y. lipolytica* samples which are cultivated with/without 1 mM CuSO_4_ for5 and 6 days. Twelve samples were included, and principal component analysis (PCA) showed that all the six samples cultivated without Cu(II) (YPD_5d, YPD_6d) are clustered into one group as “yeast-state” cells, while the samples growing with 1 mM CuSO_4_ are separated into two parts, in which the first part has all the samples cultivated with Cu(II) for 5 days (Cu_5d) and the second part has all the samples cultivated with Cu(II) for 6 days (Cu_6d) (Figure 4A). Despite that the cell morphology of Cu_5d is similar to that of YPD_5d (Figure 1B), the gene expression profile of Cu_5d is quite different from that of YPD_5d (Figure 4A). Therefore, we defined these samples to the other two groups, one is the “transition-state” cells (Cu_5d), and the other is the “hyphae-state” cells (Cu_6d). According to the PCA coordination, it seems that yeast cells first need to enter the transition state and then turn back to the hyphae state in the vertical axis (Figure 4A).

Differentially expressed genes (DEGs) were then identified by comparing the gene expression of yeast-, transition- and hyphae-state cells (Appendix A). The results showed: (1) 1174 genes were significantly changed in the transition state when compared with the yeast state (transition/yeast), in which 700 genes were upregulated and 474 genes were downregulated; (2) 1904 genes were significantly changed in the hyphae-state when compared with yeast state (hyphae/yeast), in which 1014 genes were upregulated and 950 genes were downregulated; (3) 2048 genes were significantly changed in hyphae-state when compared with transition state (hyphae/transition), in which 970 genes were upregulated and 1078 genes were downregulated (Figure 4B). In sum, 3292 unique genes were involved in the yeast-to-hypha transition, accounting for more than 40% of all the annotated genes in *Y. lipolytica*. In addition, consistently changed genes, which are DEGs shared in all three comparisons, only take a small proportion (101 genes), and the number was less than that of condition-specific DEGs, which are 355, 508, and 636 in transition/yeast, hyphae/yeast, and hyphae/transition, respectively (Figure 4B,D). Furthermore, shared DEGs of every two conditions were divided into four quadrants (Figure 4C,E,F), and the results showed that most of the DEGs shared by hyphae/yeast and hyphae/transition were consistently expressed, in which 576 genes (Q1) were both upregulated and 486 genes (Q3) were both downregulated (Figure 4C). Similar results were obtained when the DEGs shared by hyphae/yeast and transition/yeast were assigned to four quadrants (Figure 4E). By contrast, the DEGs shared by hyphae/transition and transition/yeast were mostly fallen into the second quadrant (Q2, 230 genes) and the fourth quadrant (Q4, 147 genes), suggesting that a profound gene expression turnover is presented during the dimorphic transition in *Y. lipolytica* (Figure 4F). 

To validate the reliability of the transcriptome analysis, 20 genes related to ion transport, cell division, and transcriptional regulation from transcriptomic data were selected for the qPCR analysis (Appendix A). The real-time RT-PCR data showed similar expression patterns with RNA-Seq data, suggesting that the RNA-Seq data have high reliability. Among all these ~3200 unique DEGs, more than 45% of them have not been annotated by the KOG database, and are, therefore, hypothetical genes with unknown functions. The rest of the DEGs can then be grouped into four KOG groups and ~25 classes (Table 2). In the KOG group of cellular processes and signaling, the class of posttranslational modification, protein turnover, chaperones, the class of signal transduction mechanisms, and the class of intracellular trafficking, secretion, and vesicular transport were the top three influenced KOG classes, as revealed by the count of unique genes in sum. In the KOG group of information storage and processing, the class of translation, ribosomal structure and biogenesis, the class of replication, recombination, and repair, and the class of transcription were the top three influenced KOG classes. In the KOG group of metabolisms, the most influenced KOG classes are lipid transport and metabolism, followed by energy production and conversion, amino acid transport and metabolism, and carbohydrate transport and metabolism. DEGs in these three KOG groups draw a big picture of transcriptomic changes in *Y. lipolytica*. Notably, multiple KOG classes showed uneven gene expression patterns during the yeast-to-hypha transition. For example, the upregulated genes in the class of cell wall/membrane/envelope biogenesis are more than two-fold as much as downregulated genes in it. A similar situation can also be found in the class of cytoskeleton, in which the number of upregulated genes is much more than the number of downregulated ones. The results indicated that the two classes, which are both related to cell structure, are generally upregulated in the yeast-to-hypha transition. In general, uneven gene expression patterns are found to have more upregulated genes than downregulated ones. Among them, the most significant result is in the class of translation, ribosomal structure, and biogenesis, in which class the number of upregulated genes is more than ten-fold as much as downregulated genes, and this no doubt highlight the importance of transcriptional regulation in the progress of the dimorphic transition. However, there are a few exceptions. For example, downregulated genes in the class of replication, recombination, and repair are much more than upregulated genes in it, which is consistent with our previous hypothesis that Cu(II) causes a detrimental effect on cells.

### 2.5. Functional Enrichment of Gene Expression Changes

To further elucidate the biological functions of the gene expression changes, functional enrichment analyses were performed with KEGG pathways and the GSEA method. Several KEGG pathways were found to be suppressed or activated during the dimorphic transition, and the results can be divided into three categories, which are related to signaling pathways, metabolism, and others (Figure 5).

First of all, these signaling pathways, including chemokine/Rap1/Ras/phospholipase D/VEGF/FoxO/calcium/cGMP-PKG/mTOR signaling pathways and longevity regulating pathway (Figure 6, top-down, from the first row to mTOR signaling pathway), are generally suppressed in the first stage of dimorphic transition (transition/yeast, or from yeast state to transition state), and activated in the second stage (hyphae/transition, or from transition state to hyphae state). Notably, gap junction, which contains intercellular channels that allow direct communication between the cytosolic compartments of adjacent cells, and tight junction, which is essential for establishing a selectively permeable barrier to diffusion through the paracellular space between neighboring cells, were activated in the second stage, enables multicellular behaviors of signaling in responding Cu(II) treatment. Like FoxO/mTOR signaling pathway, oxidative phosphorylation and thermogenesis were firstly suppressed in the first stage and then activated in the second stage. All the above-mentioned pathways are close to the cell cycle in yeast, as measured by the Jaccard similarity index of pathway-contained genes. Secondly, four metabolic pathways, which are biosynthesis of amino acids, carbon metabolism, citrate cycle, and cysteine and methionine metabolism, were consistently suppressed in stage 1 (Figure 5). By contrast, six pathways (from amino sugar and nucleotide sugar metabolism to aminoacyl-tRNA biosynthesis) were consistently activated in stage 2 (Figure 5). In addition, fatty acid degradation was activated in stage 1. Metabolism of xenobiotics by cytochrome P450 was activated, while glyoxylate and dicarboxylate metabolism was suppressed in stage 2 (Figure 5). Thirdly, the rest four pathways (from various types of N-glycan biosynthesis to ribosome) were conditionally expressed in stage 1 and stage 2.

Interestingly, GSEA revealed the greatest number of significantly changed pathways in stage 2, followed by stage 1, but not in stage 1 + stage 2 (hyphae/yeast, or from yeast to hyphae). Although the number of differentially changes genes in hyphae/yeast was as much as that in hyphae/transition (Figure 5), only four pathways were found activated in the hyphae/yeast comparison. In addition, a turnover of six pathways was found, of which five were firstly suppressed (FoxO signaling pathway, oxidative phosphorylation, thermogenesis, mTOR signaling pathway, and ribosome) and one (proteosome) was firstly activated, respectively (Figure 5). 

### 2.6. Novel Genes Involved in Cu-Induced Dimorphic Transition

The comparison of yeast and hyphal cell transcriptomes revealed that the expression of a large number of genes was changed in the yeast-to-hypha transition of *Y. lipolytica*. It is noteworthy that some genes showed extremely significant differences and the expression of genes that were reported to regulate dimorphic transition in *Y. lipolytica* under nutrition deficiency and some other conditions were also changed, such as *MHY1* and its direct target genes, and some cell wall proteins encoding-genes (Table 3). To validate the differentially expressed genes obtained, we constructed Δ*mhy1* and *YALI1_D11653g*-overexpressing mutant strain, and then found that Δ*mhy1* maintained yeast-form growth in the presence of 0, 1 mM CuSO_4_ and did not form hyphae compared with the wild type, while *YALI1_D11653g*-overexpressing strain promoted filamentation (Figure 6A). These results suggesting that Mhy1 plays an important role in yeast-to-hypha transition of *Y. lipolytica*, and YALI1_D11653g promotes filamentation, as described previously [39]. 

Furthermore, we selected 33 highly differentially expressed genes related to ion transport, cell division, and transcriptional regulation from transcriptomic data for overexpression (Figure 6A and Appendix A), and found that four of them, which are *YALI1_B07500g*, *YALI1_C12900g*, *YALI1_E04033g*, and *YALI1_F29317g*, promoted filamentation in the presence of 0/1 mM Cu(II) (Figure 6A). *YALI1_B07500g* encodes a protein of 351 aa. that shares sequence similarity (42.4% identity) with *S. cerevisiae* zinc finger protein STP3, which is involved in pre-tRNA splicing and uptake of branched-chain amino acids. *YALI1_C12900g* encodes a protein of 269 aa. that shares sequence similarity (31.5% identity) with *S. cerevisiae* suppressor protein STM1, which binds specifically G4 quadruplex and purine motif triplex nucleic acid structures. *YALI1_E04033g* encodes a protein of 147 a.a. that shares sequence similarity (62.1% identity) with *Schizosaccharomyces pombe* myosin regulatory light chain cdc4, which is involved in cytokinesis and required for the formation and function of the contractile ring. *YALI1_F29317g* encodes a protein of 111 a.a. that shares sequence similarity (38.8% identity) with *S. cerevisiae* transcription initiation factor IIA subunit 2, which is a component of the transcription machinery of RNA polymerase II and plays an important role in transcriptional activation. As shown in Figure 6B, all these genes were overexpressed successfully, of which about 75% were induced more than four-fold (log2 fold change > 2). Thus, our results show that the overexpression of four novel genes, which are *YALI1_F29317g*, *YALI1_B07500g*, *YALI1_E04033g*, and *YALI1_C12900g*, can promote filamentation of *Y. lipolytica*.

## 3. Discussion

*Y. lipolytica* is considered an ideal platform for the production of industrial feedstock such as biofuels and oil chemicals, and it is one of the most important chassis organisms in synthetic biology [47,48]. This is a dimorphic fungus able to grow in metal-ion-contaminated environments, and can also be affected by the metal ions to undergo a yeast-to-hypha transition [11]. However, the mechanism of metal-induced dimorphic transition in *Y. lipolytica* remained unclear. In the current study, we revealed that Cu(II) could significantly promote the yeast-to-hypha transition of *Y. lipolytica*. Moreover, we have drawn a picture of Cu(II)-induced dimorphic transition regulatory pathways by RNA-seq analysis. Furthermore, we identified four novel genes that can influence the dimorphism of *Y. lipolytica*. These findings provided insights into the understanding and engineering of the dimorphic transition of this important organism.

Here, we detected Cu(II) accumulation of *Y. lipolytica* before and after hyphae formation and the results showed that the total and intracellular Cu(II) accumulation drastically decreased after hyphae formation. This change may be a strategy for *Y. lipolytica* to respond to metal toxicity. The morphological change from the oval-shaped yeast form to the hyphae is beneficial to biomass accumulation and is a common response to environmental stress by lowering the surface/volume ratio of *Y. lipolytica* [47]. This view was also verified by the results of trypan blue staining, which showed that yeast-to-hypha transition in *Y. lipolytica* is beneficial for its survival in the presence of Cu(II) (Figure 3B). Interestingly, several genes involved in intracellular Cu(II) transport were up-regulated in the second stage, including *YALI1_B23771g*, *YALI1_C05880g*, *YALI1_C28396g*, suggesting that cells showed increased efflux of Cu(II) (Table 3). It is believed that this is beneficial to the survival of hyphal cells. The increment of cell adsorption of Cu(II) is lower than the increment of biomass, therefore, a decrease in the total and intracellular Cu(II) accumulation was observed.

Given that Cu(II) is not consumed during the dimorphic transition, we proposed that Cu(II) acts as a signaling molecule to induce a series of changes and then result in the yeast-to-hyphal dimorphic transition. In consistent to this hypothesis, our transcriptome analysis indicated that many signaling pathways were activated in the second stage (hyphae/transition, or from transition state to hyphae state), including chemokine/Rap1/Ras/phospholipase D/VEGF/FoxO/calcium/cGMP-PKG/mTOR signaling pathways and longevity regulating pathway. Additionally, a gap junction was also activated in the second stage, suggesting that the cell–cell communication connection is closed and can rapidly and reversibly promote the synergistic responses of neighboring cells to external signals. Thus, it seems that Cu(II) functions as a signaling molecule in the dimorphic transition. However, how Cu(II) transmits signals is still unknown.

Previous studies have revealed reactive oxygen species (ROS) can act as signaling molecules and promote fungal differentiation [49,50,51]. In *C. albicans*, the formation of hyphae was shown to be associated with a significant increase in ROS formation [49]. In the RNA-Seq data sets, the genes *YALI1_E14988g* and *YALI1_B11983g*, which encode superoxide dismutase (SOD) and are involved in the catalysis of O^2−^ were upregulated, and the genes *YALI1_C13106g* and *YALI1_D01404g*, which encode oxidoreductases, were upregulated as well (Table 3). These genes might be involved in the resistance to peroxide, and the results suggest that excessive copper may lead to the generation of ROS, such as superoxide anions (O^2−^), and cause cellular damage in vivo. Therefore, Cu(II)-induced dimorphic transition may also be related to ROS generation.

The transcriptomic data analysis showed that *Y. lipolytica* underwent a dimorphic transition that has three distinct states: yeast state, transition state, and hyphae state, in which the gene expressions between them are quite different from each other. The gene expression pattern of the “transition state” was significantly changed, compared with the yeast state, although the cells were still in yeast form. This is a critical step for yeast cells to transform into hyphal state. After that, more than 25% of all the annotated genes in *Y. lipolytica* were significantly up/down-regulated (hyphae/transition). The finding of the transition state suggests that some extremely complex changes occurred during the dimorphic transition. During the transition, the gene expressions of several genes were even turned over. This observation set a good example of how the cell fate of eukaryotic organisms is regulated.

During the Cu(II)-induced dimorphic transition in *Y. lipolytica*, we could find the participation of many recently reported genes are involved, including *YlGPI7*, *YlCDC25*, *YlRAC1*, *YlTEC1*, *YlRSR1*, and *FTS2* (Table 3) [41]. These genes should play a common role in the yeast-to-hypha transition of *Y. lipolytica*. Moreover, the MAPK and cAMP-PKA signaling pathway is also found to be involved in the dimorphic transition in *Y. lipolytica* under nutrient-deficient conditions [25,52]. However, the gene expression changes related to MAPK and cAMP-PKA signaling pathway, including *YlHOG1*, *YlSTE11*, *YlKSS1*, and *YlTPK1*, are not found by the transcriptome analysis [53,54], suggesting that these pathways may have little effect on the regulation of Cu(II)-induced dimorphic transition.

Finally, we identify multiple novel genes involved in filamentous growth—*YALI1_F29317g*, *YALI1_B07500g*, *YALI1_E04033g*, and *YALI1_C12900g*. The overexpression of these four genes could promote yeast-to-hyphal transition in the absence of Cu(II), suggesting these genes played a critical role in regulating the dimorphic transition of *Y. lipolytica*. And it’s worth further investigations into what are the particular roles of these four genes in regulating dimorphic transition. 

## 4. Conclusions

In summary, Cu(II) significantly promoted yeast-to-hypha transition in *Y. lipolytica*, and the rate of hyphae formation was positively correlated with the Cu(II) concentration in a certain range. The total and intracellular Cu(II) accumulation of cells decreased after hyphae formation. In contrast, the intracellular Cu(II) accumulation was evidently increased before hyphae formation. In addition, Cu(II) had a toxic effect on the cells, which resulted in the reduction of the viability and TLL activity, and yeast-to-hypha transition in *Y. lipolytica* is beneficial for its survival in the presence of Cu(II). Furthermore, the regulatory mechanisms of Cu(II)-induced dimorphic transition have been revealed through RNA-seq analysis. Moreover, we successfully identified four novel genes that could significantly promote yeast-to-hyphal transition in the absence of Cu(II). Ultimately, the present results revealed the effect of Cu(II) on the morphology and physiological function of *Y. lipolytica*.

## 5. Materials and Methods

### 5.1. Strains, Plasmids, and Culture Conditions

*Y. lipolytica* strains and plasmids used in this study are listed in Appendix A. The *Y. lipolytica* strain W29 (CLIB89, *MATa* WT) was maintained by this laboratory and used as the wild-type strain, and strain Po1f (uracil, leucine double deletion) was used for constructing overexpressing strains. The *E. coli* strains TOP10 and TOP10 F’ (Shanghai Weidi Biotechnology Co., Ltd., Shanghai, China) were employed for plasmid amplification.

To initialize a *Y. lipolytica* culture, glycerol stock was first streaked onto a YPD plate and cultivated at 28 °C overnight. After that, a single colony was picked and inoculated into a 20 mL bottle containing 5 mL YPD liquid medium (10 g yeast extract, 20 g peptone per liter, 2% glucose, pH 6.5) at 28 °C and 200 rpm for 20 h. Then, 1 mL of the culture was transferred into a 250 mL Erlenmeyer flask containing 100 mL YPD liquid medium and cultivated under the same condition for another 36 h (seed culture, OD_600_ is approximately 5). Then, 1mL of seed culture was inoculated into a 250 mL flask containing 100 mL YPD liquid medium, and shaken with 200 rpm at 28 °C. Unless otherwise stated, 2mM CuSO_4_ (Sinopharm Chemical Reagent Co., Ltd., China, pH 6.5) was supplemented to induce dimorphic transition if needed. The *E. coli* strains were grown in LB liquid medium (5 g yeast extract, 10 g tryptone, and 10 g NaCl per liter, pH 7.0) with ampicillin (100 μg/mL, if needed) at 37 °C and used for plasmid construction.

### 5.2. Construction of Overexpressing Strain

In order to construct overexpressing strains, a series of recombinant plasmids were cloned and transformed into the Polf strain of *Y. lipolytica*. For instance, to overexpress *YALI1_D11653g*, two pairs of primers N-D11653g-F/NsiI-D11653-R and NsiI-D11653-F/C-D11653g-R (see Appendix A) were used to amplify the *YALI1_D11653g* ORF sequence by overlapping PCR for the synonymous mutation to remove the NsiI restriction site. Then, after digestion with Not I & ClaI, the ROL sequence of pUAxp7166-ROL (hp4d promoter and XPR2 terminator expression cassette, URA3 selection marker. Designed by Dr. Jiao Liangcheng [55]) was replaced by homologous recombination with the amplified *YALI1_D11653g* ORF sequence, yielding pUAxp7166-D11653g. Subsequently, the transformation of *Y. lipolytica* Po1f with the plasmid digested by NsiI was conducted to generate the overexpressing strain as described previously [56]. Likewise, 33 highly differentially expressed genes revealed by transcriptome analysis were selected and the corresponding overexpressing strains were constructed in similar manners with corresponding primers (Appendix A).

### 5.3. Construction of MHY1 Knockout Strain

Δ*mhy1* strain was constructed in *Y. lipolytica* strain Po1f using CRISPR-Cas9 knockout. A 20 bp sgRNA sequence, which was selected from the ORF sequence of *MHY1*, was amplified by PCR, and then inserted into the pCRISPRyl plasmid linearized by BlnI with homologous recombination. The resulting pCRISPRyl-mhy1 was transformed into strain Po1f by the lithium acetate method. Ura+ transformants were selected and sequenced to identify the correct Δ*mhy1* mutant clones. The primers used in this experiment are listed in Appendix A.

### 5.4. Measurement of Cell Morphologies, Hyphal Quantification, and Dry Cell Weight

*Y. lipolytica* cells were grown under the conditions mentioned above. For morphology observation, 10 μL cultures of each sample were loaded onto a slide and covered with a 20 × 20 mm^2^ cover slip. The cell morphology images were taken every 1 day using a differential interference contrast (DIC) microscope (NE910, China). For quantification analysis, the samples were first diluted 10-fold, and 10 μL of diluted culture was loaded onto a glass slide and covered with a 20 × 20 mm^2^ cover slip. We randomly selected 20–30 fields under the microscope view with a 40× objective and 10× eyepiece, and counted about 600 cells (hyphal cells + yeast-form cells) with the assistance of the eyepiece ruler (cell length can be obtained by referring to the ruler) and live video stream. During this process, cells longer than 20 μm were scored as hyphal cells, while the mother cell and the bud associated with the mother cell were counted as one cell. The experiments were performed in triplicates. In order to measure the dry cell weight, 5 mL cells were harvested in 10 mL centrifuge tubes and washed twice with distilled water by centrifugation. The cells were weighed after heating for 48 h at 100 °C (dry cell weight, DCW).

### 5.5. Assay of Copper Ion Accumulation

*Y. lipolytic*a cells were grown in YPD liquid medium supplemented with 0~2 mM CuSO_4_ at 28 °C for 4–8 days. For total Cu(II) accumulation determination, cells were collected by centrifugation and washed three times with deionized water. For intracellular Cu(II) accumulation determination, cells were collected by centrifugation and washed three times with 2 mM EDTA. Then, the collections were digested with 6 M HNO_3_, boiled for 20 min, and filtered with 0.25 μm filters to remove organic matter. After diluting to the appropriate concentration, the Cu(II) accumulation of digested samples was quantified by atomic absorption spectrometer (iCE3300, USA) according to the manufacturer’s instructions.

### 5.6. Quantification of Cell Viability

The cell viability was determined by colony forming units (CFU) counting and trypan blue staining. For CFU counting assays, 100 μL of the sample cultures were diluted to an appropriate gradient series, and the gradients were plated on solid YPD plates. After 24 h incubation at 28 °C, plates with colonies in the range of 30–300 were finally selected for counting. Then the CFU of the sample was calculated according to colonies numbers and the related diluted concentration. For trypan blue staining, the sample cultures were diluted to the appropriate concentration (each small square of the hemocytometer contains 4–5 yeast cells) and stained with trypan blue (final concentration at 0.04%) for 4 min. The stained cultures were transferred into the hemocytometer and the dead and living cells were distinguished and counted based on dyeing or not under the ordinary light microscope (ML31, China) within 10 min. 

### 5.7. Determination of Thermomyces Lanuginosus Lipase (TLL) Activity

The TLL activity of the cell culture supernatant was measured according to a titrimetric method [57]. The reaction system consisted of 4 mL olive oil emulsion (olive oil: 2% polyvinyl alcohol = 3:1), 5 mL Tris-HCl solution (pH 8.0, 50 mM), and 1 mL suitably diluted cell culture supernatant was placed in a water bath shaker at 60 °C for 10 min, terminated with 15 mL 95% ethanol, and finally titrated with 50 mM NaOH using phenolphthalein as an indicator. The unit of lipase activity (U) is defined as the amount of the enzyme that releases 1 μmol of fatty acid per minute.

### 5.8. RNA Extraction and RNA-Seq Analysis

The *Y. lipolytica* strain W29 was grown in YPD liquid medium supplemented with 0, 1 mM CuSO_4_ at 28 °C. Cells cultured for 5 and 6 days were harvested and frozen quickly using liquid nitrogen. Three replicates were performed for each sample. The following steps including total RNA extraction, library construction, and sequencing were performed by Frasergen Information Co., Ltd. (Wuhan, China). Total RNA was extracted from samples, with the concentration and purity detected by Nanodrop 2000 and RIN value determined by Agilent 2100. A total amount of 2 μg total RNA was used for library construction and mRNA was enriched using poly-T oligo-attached magnetic beads. Qualified libraries were prepared as DNA nano-ball for sequencing, using MGI high-throughput sequencer (MGISEQ-2000). After base calling and Bcl2fastq transformation, the raw data were transformed as FASTQ format, called raw reads, which were processed and transformed into clean and high-quality data by removing reads containing adapter, low-quality sequence, and poly (N) tail using SOAPnuke software (v2.1.0.) [58]. DEGs were screened by p.adjust < 0.05 and Log2FC < −0.5 or Log2FC > 0.5. KOG classifications of genes were adopted from the JGI genome portal [59]. Gene set enrichment analysis (GSEA) was performed to identify activated and suppressed KEGG pathways in *Y. lipolytica* using the clusterProfiler package, as described previously [60,61]. Since the gene annotation of metabolic pathways in *Y. lipolytica* is out of date, we generated the KEGG annotation of genes with the KEGG mapping service before performing GSEA [62]. The annotation can be accessed/reused as a Bioconductor Orgdb object at https://github.com/gaospecial/org.Ylipolytica.W29.eg.db. Raw sequencing data has been deposited into NCBI SRA repository under the bioproject PRJNA933354.

### 5.9. qRT-PCR Analysis

qRT-PCR was performed using the AceQ qPCR SYBR Green Master Mix (Vazyme Biotech Co., Ltd., China) on the QuantStudio™ 3 system (Thermo Fisher Scientific, Waltham, MA, USA). Primers for the qRT-PCR analysis were designed using Primer3 Plus, and the information of those primers is listed in Appendix A. *ACT1*(*YALI0D08272g*) was used as the reference gene based on its relatively constant expression. Gene expression changes were calculated using the 2^−ΔΔCt^ method [63].

## Figures and Tables

**Figure 1 jof-09-00249-f001:**
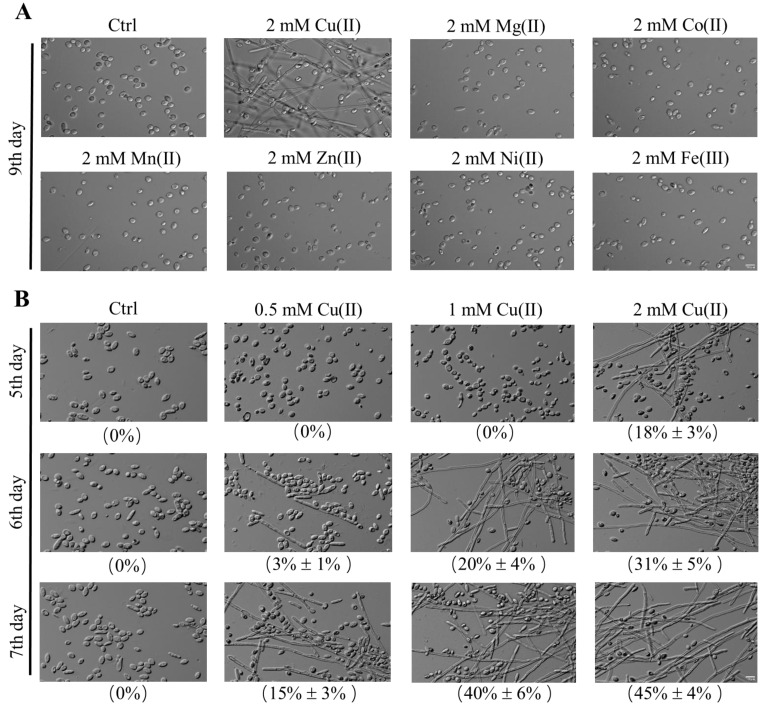
Cu(II)-induced yeast-to-hypha transition in *Y. lipolytica*. (**A**) Cells of strain W29 were grown in YPD liquid media in the absence (Ctrl) or presence of 2 mM CuSO_4_, MgSO_4_, CoCl_2_, MnSO_4_, ZnCl_2_, NiCl_2_, and FeCl_3_ at 28 °C. Pictures were taken after being grown at 28 °C for 9 days. (**B**) Cells of strain W29 were grown in YPD liquid media in the absence (Ctrl) or presence of 0.5, 1, and 2 mM CuSO_4_ at 28 °C. Pictures were taken after being grown at 28 °C for 5 to 7 days. Scale bar: 10 μm. The number in parentheses shows the percentage of cells longer than 20 μm (n > 600 cells), and the mother cell and the bud associated with the mother cells were counted as one cell. Figures are the representative results of different conditions and the average of three independent replicates was shown on the bottom (mean ± SD).

**Figure 2 jof-09-00249-f002:**
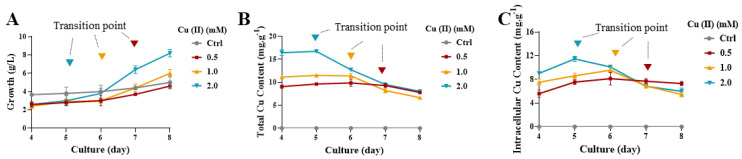
Cu(II) accumulation during Cu(II) -mediated dimorphic transition in *Y. lipolytica***.** (**A**) DCW of the *Y. lipolytica*. (**B**) Total or (**C**) intracellular Cu(II) accumulation of the *Y. lipolytica*. For (**A**–**C**): cells of the *Y. lipolytica* Strain W29 were grown in YPD liquid media in the absence (Ctrl) or presence of 0.5, 1, and 2 mM CuSO_4_ at 28 °C for 4 to 8 days. Data are presented as averages of biological triplicates (n = 3). Triangles indicate the start of dimorphic transition in different cultures, respectively.

**Figure 3 jof-09-00249-f003:**
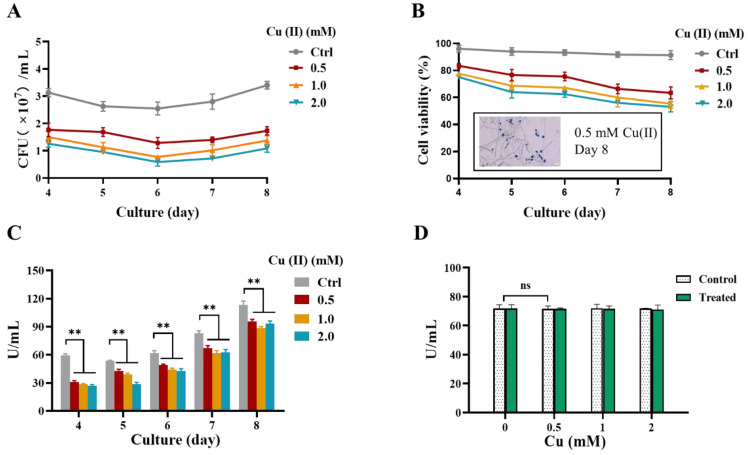
Cell viability and thermomyces lanuginosus lipase (TLL) activity during Cu(II)—mediated dimorphism in *Y. lipolytica*. (**A**) Colony-forming units of the *Y. lipolytica* strain Po1f under various conditions. (**B**) The viability of the yeast-form cells of strain Po1f is detected using trypan blue staining in which dead cells are stained in blue and live cells are transparent, then cell viability is calculated by the formula: number of live cells/(number of live cells and dead cells). Inset, a representative image of trypan blue staining of the Day 8 culture with 0.5 mM CuSO_4_, Scale bar: 10 μm. (**C**) TLL activity of the *Y. lipolytica* strain Po1f. The asterisk indicates statistically significant differences between copper-treated samples and the control (** Pr ≤ 0.01). (**D**) TLL activity of the *Y. lipolytica* strain Po1f under various conditions in vitro. The direct reaction between Cu(II) and cell-free supernatant collected on the 6th day. CuSO_4_ was added with a final concentration of 0~2 mM and made up the volume to 1 mL. The control is an equal volume of water. TLL activity was measured after being incubated at 28 °C for 6 h. For (**A**–**C**), cells of the *Y. lipolytica* were grown in YPD liquid media in the absence (Ctrl) or presence of 0.5, 1, and 2 mM CuSO_4_ at 28 °C for 4 to 8 days. Data are means ± SD, with n ≥ 3.

**Figure 4 jof-09-00249-f004:**
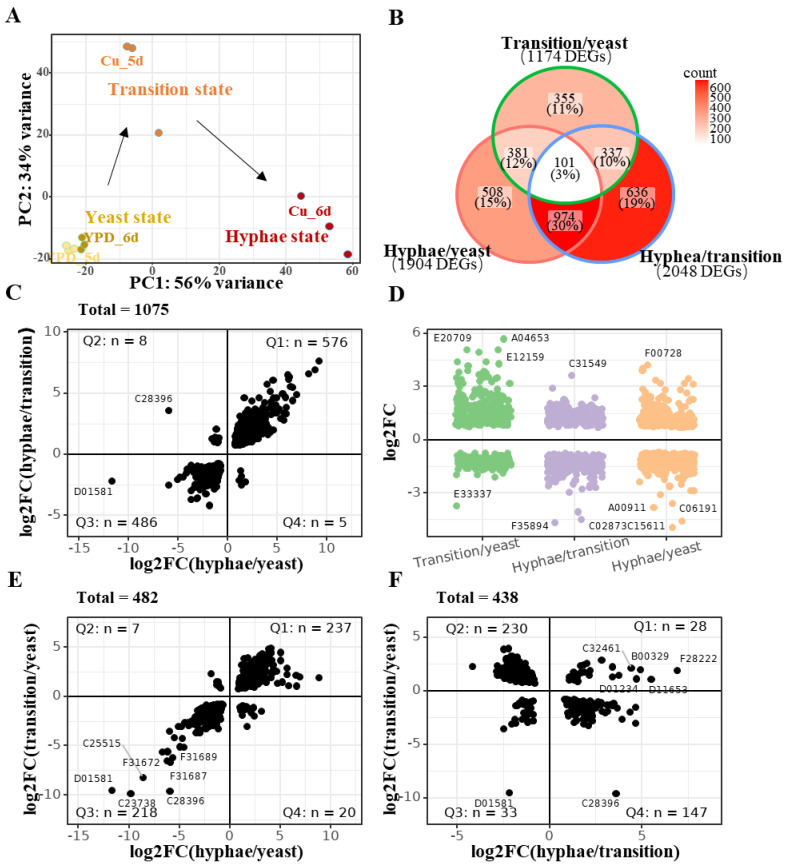
Transcriptomic changes during Cu(II)-mediated dimorphic transition in *Y. lipolytica*. (**A**) Principal component analysis (PCA) and the definition of the yeast-, transition- and hyphae-state cells. Arrows indicate the progress of the yeast-to-hypha transition can be treated as two steps. (**B**) Venn diagrams showing the overlap of genes differentially expressed under three comparisons (log2FC > 0.5, p.adj < 0.05). Counts of DEGs in each subset were given in places, and the proportions are calculated with the number of unique genes in three comparisons. Region-filling colors also indicated the variance in the number of DEGs. (**C**–**F**) Gene expression changes of DEGs. Each point represents a gene, and the adjacent text label indicates what the gene is. To save space, only the unique parts of gene model names are given in text labels. (**C**,**E**,**F**) A comparison of the consistency of the shared DEGs in three comparisons. In each subplot, gene expression changes were plotted as a point by two dimensions, which correspond to two of the three comparisons. The panel was divided into four quadrants (Q1, Q2, Q3, and Q4) by horizontal and vertical lines anchored in the zero point of two axes. If a gene was upregulated or downregulated consistently, it will appear in Q1 and Q3, respectively. By contrast, if the gene expression was turned over, it will appear in Q2 or Q4. To align the three subplots vertically or horizontally, they are placed in (**C**), (**E**), and (**F**), separately. (**D**) Comparison-specific DEGs. These genes’ expression changes are plotted by comparison group as indicated by the horizontal axis and colors.

**Figure 5 jof-09-00249-f005:**
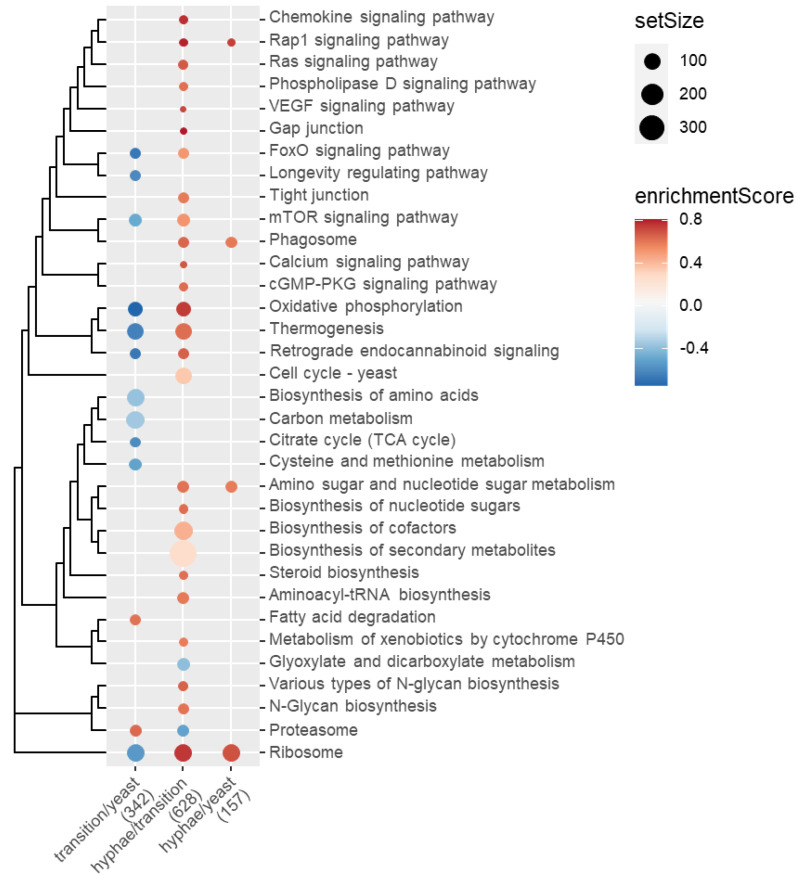
GSEA analysis of KEGG pathways in Cu(II)-mediated dimorphic transition. There are approximately 2300 genes were assigned to 400 KEGG pathways, and those significantly activated or suppressed in at least one comparison (horizontal axis) were shown, after manual curation of the raw GSEA result. These pathways were clustered by the Jaccard similarity and shown as a dendrogram (**left**). Point size indicates how many different genes were included for a pathway, and color indicates the enrichment score, of which a negative value implies the pathway is suppressed, and a positive value implies the pathway is activated, respectively.

**Figure 6 jof-09-00249-f006:**
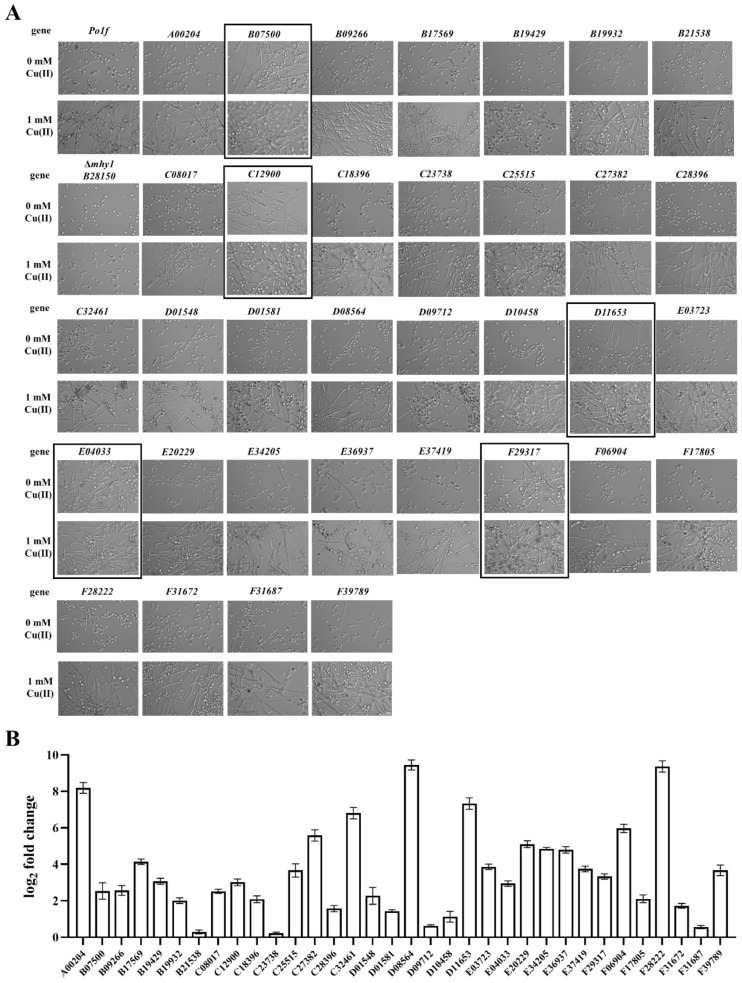
Screening of novel regulators that are essential in Cu—induced dimorphic transition. (**A**) The screening was performed with a series of gene overexpressing strains, which are sorted alphabetically in this figure. *YALI1_B07500g*, *YALI1_C12900g*, *YALI1_D11653g*, *YALI1_E04033g*, and *YALI1_F29317g*, promote filamentation upon overexpression (highlighted by solid boxes). Cells of the wild type of Po1f strain (the first one in the first row), and Δ*mhy1* (encoded by *YALI1_B28150*) mutant (the first one in the second row), which is a known essential gene in dimorphic transition, were used as controls. Pictures were taken after being grown at 28 °C for 8 days. Scale bar: 10 μm. (**B**) Relative expression levels of the corresponding genes in the 34 overexpressed strains with Po1f strain as a control. All samples were collected on the 8th day and *ACT1*(*YALI0D08272g*) was used as the reference gene to normalize the gene expression in fluorescent real-time RT-PCR. Data are presented as averages of biological triplicates (n = 3). For (**A**,**B**), all strains were grown in YPD liquid media buffered at 0, 1 mM CuSO_4_ at 28 °C.

**Table 1 jof-09-00249-t001:** Exploring the minimum and maximum concentrations of Cu(II)-induced filamentation in *Y. lipolytica*.

Cu(mM) Day	0	0.001	0.002	0.003	0.004	0.005	0.05	0.5	1	2	4	6	7	8
0	×	×	×	×	×	×	×	×	×	×	×	×	×	×
1	×	×	×	×	×	×	×	×	×	×	×	×	×	×
2	×	×	×	×	×	×	×	×	×	×	×	×	×	×
3	×	×	×	×	×	×	×	×	×	×	×	√	×	×
4	×	×	×	×	×	×	×	×	×	×	√	√	×	×
5	×	×	×	×	×	×	×	×	×	√	√	√	×	×
6	×	×	×	×	×	×	×	×	√	√	√	√	×	×
7	×	×	×	×	×	×	×	√	√	√	√	√	×	×
8	×	×	×	×	×	×	√	√	√	√	√	√	×	×
9	×	×	×	×	√	√	√	√	√	√	√	√	×	×
10	×	×	×	√	√	√	√	√	√	√	√	√	×	×
11	×	×	×	√	√	√	√	√	√	√	√	√	×	×
12	×	×	×	√	√	√	√	√	√	√	√	√	×	×

×: no hyphae formed, √: hyphae formed. Cells of the strain W29 were grown in YPD liquid media in the presence of 0 to 8 mM CuSO_4_ at 28 °C and its morphology was observed at 0 to 12 days.

**Table 2 jof-09-00249-t002:** Overall gene expression changes during the yeast-to-hypha transition.

KOG Group	KOG Class	Yeast to Transition (up/dn)	Transition to Hyphae (up/dn)	Yeast to Hyphae (up/dn)	Total (Unique)
CELLULAR PROCESSES AND SIGNALING	Cell wall/membrane/envelope biogenesis	2/2	11/5 ^#^	13/3 ^#^	24
	Cytoskeleton	2/5	32/3 ^#^	29/4 ^#^	44
	Defense mechanisms	1/3	6/9	3/4	16
	Extracellular structures	0/3	5/1	3/3	8
	Intracellular trafficking, secretion, and vesicular transport	13/9	48/22 ^#^	61/17 ^#^	114
	Nuclear structure	2/1	4/4	4/0	10
	Posttranslational modification, protein turnover, chaperones	71/24 ^#^	52/94	63/53	224
	Signal transduction mechanisms	11/15	54/32	49/39	128
INFORMATION STORAGE AND PROCESSING	Chromatin structure and dynamics	6/1 ^#^	2/10 ^#^	6/3	21
	Replication, recombination and repair	15/9	12/43 ^#^	10/27 ^#^	72
	RNA processing and modification	7/3	4/10 ^#^	10/10	37
	Transcription	6/5	13/26	18/22	62
	Translation, ribosomal structure and biogenesis	4/7	95/7 ^#^	78/7 ^#^	122
METABOLISM	Amino acid transport and metabolism	25/21	40/38	35/34	120
	Carbohydrate transport and metabolism	23/18	34/26	35/28	102
	Cell cycle control, cell division, chromosome partitioning	16/8	29/16	28/11 ^#^	68
	Coenzyme transport and metabolism	5/10	21/4 ^#^	12/8	36
	Energy production and conversion	17/35 ^#^	62/30 ^#^	27/27	124
	Inorganic ion transport and metabolism	16/23	14/11	18/30	64
	Lipid transport and metabolism	44/10 ^#^	28/60 ^#^	51/36	140
	Nucleotide transport and metabolism	4/6	12/7	8/7	28
	Secondary metabolites biosynthesis, transport and catabolism	21/19	13/18	23/28	69
CELLULAR PROCESSES AND SIGNALING	Cell motility		0/1	0/1	1
POORLY CHARACTERIZED	Function unknown	19/17	33/35	25/30	104
	General function prediction only	43/45	95/102	82/95	286

^#^ Uneven numbers of DEGs were found in this class, as the upregulated genes are more than two-fold of downregulated genes, or *vice versa*.

**Table 3 jof-09-00249-t003:** Genes involved in the dimorphic transition of *Y. lipolytica*.

GeneID	Hyphae /Yeast ^1^	Transition /Yeast ^2^	Hyphae /Transition ^3^	Description of the Encoded Protein ^4^	Ref. ^5^
YALI1_B07500g	−1.237			C2H2-type domain-containing protein	this study
YALI1_C12900g	1.216		1.757	HABP4_PAI-RBP1 domain-containing protein	this study
YALI1_E04033g	3.289		3.228	EF-hand protein	this study
YALI1_F29317g	1.079		1.312	Transcription initiation factor IIA subunit 2	this study
YALI1_B11983g		−1.579	1.589	Superoxide dismutase	this study
YALI1_B19932g	2.218		2.915	Thiol-specific antioxidant	this study
YALI1_B23771g			1.928	Similar to copper transport protein, ATX1	this study
YALI1_C05880g		−1.998	2.018	Similar to copper transport protein, SMF1	this study
YALI1_C13106g		−1.537	1.48	NADPH-dependent diflavin oxidoreductase 1	this study
YALI1_C28396g	−5.962	−9.551	3.589	Similar to copper transport protein, CTR1	this study
YALI1_D01404g	2.539		2.311	Oxidoreductase activity	this study
YALI1_E14988g	2.407	1.352	1.055	Superoxide dismutase [Cu-Zn]	this study
YALI1_B13328g	2.738		4.643	Downstream target genes of Mhy1	[40]
YALI1_B17773g	3.689		3.624	BHLH domain-containing protein	[41]
YALI1_B28150g	1.673		3.112	Mhy1, C2H2-type zinc finger protein	[40,42]
YALI1_C15610g	2.436	−1.932	4.368	YlRim101- and Mhy1-coregulated gene	[39]
YALI1_C21578g	6.252		6.359	Downstream target genes of Mhy1	[40]
YALI1_C32352g	6.352		5.541	Downstream target genes of Mhy1	[40]
YALI1_D06131g	4.353		4.323	1,3-beta-glucanosyltransferase, *Yl*Phr1	[39]
YALI1_D07729g	2.306		2.106	1,3-beta-glucanosyltransferase, *Yl*Phr2	[39]
YALI1_D11653g	6.574	1.078	5.496	Similar to the *S. cerevisiae* a-agglutinin Aga1	[39]
YALI1_D18089g		1.424		Zn(2)-C6 domain-containing protein	[41]
YALI1_D34684g	1.137		1.243	YlGpi7	[43]
YALI1_E15718g			1.449	YlCdc25	[43]
YALI1_E26257g	1.681	−1.575	3.929	Cell wall protein precursor YlCWP1	[39]
YALI1_E27731g	−0.861			Fts2, C2H2-type zinc finger protein	[41]
YALI1_E30639g	1.257		1.48	YlRac1	[44]
YALI1_F20378g		−1.551		YITec1	[45]
YALI1_F25382g	4.347		3.870	Similar to *S. cerevisiae* flocculin Flo11	[39]
YALI1_F30395g	1.888	2.849	−1.888	YlRsr1	[46]

Notes: ^1, 2, 3^ Log2 fold change of gene expressions in difference comparisons. ^4^ the predicted function of gene product. ^5^ References.

## Data Availability

Raw sequencing data has been deposited into NCBI SRA repository under the bioproject PRJNA933354. The genome annotation of *Y. lipolytica* used in this study can be accessed/reused as a Bioconductor Orgdb object at https://github.com/gaospecial/org.Ylipolytica.W29.eg.db (accessed on 14 May 2022). Appendix A can be accessed in publisher’s website.

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
