# Peer review of "Copper Ion Mediates Yeast-to-Hypha Transition in *Yarrowia lipolytica"

_jof, 2023, doi:10.3390/jof9020249_

Round 1
Reviewer 1 Report (Previous Reviewer 3)
Minor comment:
Line 465: despite the authors acknowledge there was a typo, and they did use YPD medium containing peptone and not tryptone for Yarrowia lipolytica cells growth, they still cite tryptone in the revised manuscript – this must be corrected.
Author Response
Reviewer 1
- Line 465: despite the authors acknowledge there was a typo, and they did use YPD medium containing peptone and not tryptone for Yarrowia lipolytica cells growth, they still cite tryptone in the revised manuscript – this must be corrected.
Response:
Thank you for pointing out this problem. We have corrected tryptone to peptone in the revised manuscript. Besides, we also revised the formula of LB medium, in which tryptone was wrongly given as peptone in the former version. In fact, predefined culture medium was used throughout this study so that we are actually not familiar with the contents very well. Nonetheless, we apologize for this fault. The revised contents are as follows:
“To initialize a Y. lipolytica culture, 1 mL of glycerol stock was transferred to 100 mL YPD liquid medium (10 g yeast extract, 20 g peptone per liter, 2% glucose, pH 6.5) and cultured at 200 rpm and 28°C for 36 h, generating the seed culture (OD600 is approximately 5). Then, 1mL of seed culture was inoculated into a 250 mL flask containing 100 mL YPD liquid medium, and shaken with 200 rpm at 28℃. Unless otherwise stated, CuSO4 (Sinopharm Chemical Reagent Co., Ltd, China) was supplemented to induce dimorphic transition if needed. The E. coli strains were grown in LB liquid medium (5 g yeast extract, 10 g tryptone, and 10 g NaCl per liter, pH 7.0) with ampicillin (100 μg/mL, if needed) at 37℃ and used for plasmid construction.”
Reviewer 2 Report (Previous Reviewer 2)
The authors have addressed my and the other reviewers concerns. I have no further suggestions for improvement.
Author Response
Thanks to your critical reading, helpful comments, and constructive suggestions.
Reviewer 3 Report (Previous Reviewer 1)
The revised manuscript replies only very partially to the comments. The experimental design is still controversial to my understanding. First of all, the authors claimed that after 4 days, cells reach the stationary phase, meaning that the study concerns the dimorphic transition of cell in “resting cells” condition and not during growth. Because we do not have the growth kinetic before 4 days, we can not be sure that all the cells are in the same stage in between the different culture conditions.
The authors claims that the cells do not encounter other stress than the one from the Cu(II), or if the cells are in stationary phase that means that the cells are in presence of at least a limiting elements (in the medium or oxygen level) or in presence of an inhibitor which would be a non-controlled stressful condition. To me it is more like a multi-stress approach than a specific Cu(II) study which make the interpretation of data much more complicated.
We still don’t know, how was determine the % of filamentation?... how many images have been analysis, on how many localizations on the glass slide, how many cells have been counted?... so many missing information necessary to be confident in the presented data.
Same information on the culture condition. What does mean “1 ml of the seed culture was used to inoculate 100ml of YPD”?.... what was the seed culture (from colony, or from bulk and in which conditions) ?...
The initial conditions of the culture are so important in order to characterize morphology that it has to be well-standardized especially if different culture conditions are tested. Because I have no confidence in the experimental design, I do have some issues to be convinced by the data and especially the one on the RNAseq.
Still not convinced by most of the response of the comments.
Round 2
Reviewer 3 Report (Previous Reviewer 1)
Still not convinced by the experimental design and therefore not convinced by the work, the data and interpretation. Sorry for that.
Author Response
Please see the attachment.

This manuscript is a resubmission of an earlier submission. The following is a list of the peer review reports and author responses from that submission.
Round 1
Reviewer 1 Report
To my opinion, this article cannot be published without supplying much more data and especially largely improving the experimental design or at least give the proof that the experimental set-up allows to draw the conclusion of the authors.
To my opinion, the experimental set-up implemented and described can absolutely not lead to draw the conclusions. No confidence at all in the described results and especially not with the “RNAseq” approach. RNAseq must be done on well controlled conditions and it seems to me that it is not the case.
The very critical issues:
1) There is absolutely no description of the way that the cultures were done. It is just precised that the cells were grown in a specific medium at 28°C. No info on the protocols (tube, erlen, reactor, ……etc). What about the pH, the quantity of dissolved oxygen… nothing is precised. Dimorphic transition is very well dependant on culture conditions and environmental parameters and nothing is precised on the way the cultures were run.
2) Furthermore, the authors must update their literature review because it is missing numerous papers on Y. lipolytica and dimorphic transition. A lot more article have been published since 2000-2010…
3) The first sample were taken after 4 days but the growth rate of Y. lipolytica in rich medium is about 0.26-0.28h-1, so the growth was done way before the 4th day of culture. Was-it a study on “resting cells” but in that case there is no more growth and the problematic of filamentation is way different. What does happen after 4 days ?... what is the point to go up to 12 days ?
4) It seems that the growth reaches up to 2-4 gL-1 but we don’t even now how it was determined, and if it is the case that it is sure that pO2 and pH must change drastically and other factors that metals could be responsible of filamentation.
5) Quantity of cells used for intracellular Cu?... it has to be precised the quantity of cells taken for the extraction.
6) Absolutely no description of the protocol to characterize the filamentation, it is just written that “cell morphology image were taken”: What does it mean?..a complete description of the protocol has to be precised (the type of equipment, the concentration of the sample, the dilution, the size of the glass surface used for analyses….)
7) The authors talked about metabolome (Line 386): Where is the metabolome study ?...
Reviewer 2 Report
General Impression
The authors present a well-written summary of their studies on the effect of copper on morphology and gene expression in the industrially important dimorphic yeast Yarrowia lipolytica. The authors demonstrate that exposure to sublethal concentrations of CU(II) triggers the yeast-to-hypha transition and effects changes in gene expression that are consistent with the transition process. In addition to confirming the role of well-known factors in the morphological transition, the data allow the authors to identify novel hyphae-promoting factors. A strength of this manuscript is that the authors actually follow up on the hypotheses suggested by the RNA sequencing data. In the last and likely the most labor-intensive process of the study, the authors show that overexpression of 5 copper-upregulated genes induce hyphae-formation. The general impression is of a well-designed and well-executed study that draws on a broad array of methods to provide insights into an important biological process; in this, the study clearly merits publication.
Strengths
The strength of the manuscripts lies in the selection and execution of a number of techniques that give a comprehensive view of copper effects on Yarrowia: The chosen studies of morphology, toxicology, gene expression and mutant phenotypes combine well into a comprehensive story.
Weaknesses
It is not clear if the overexpression of the 33 differentially expressed genes (shown in figure 6) has been verified by RNA analysis. Otherwise, it could be that the failure of 28 of the candidates to induce filamentation is due to a lack of expression of the construct.
Reviewer 3 Report
The authors investigated the effects of copper sulphate on Yarrowia lipolytica. The emphasis is on yeast-to-hypha transition and the effects on transcriptional response. Authors show that copper induce filamentation, changes in gene expression, copper accumulation differ in yeast and hyphae form. Authors used RNAseq technology to characterize major molecular changes that are associated with the morphogenetic switch. Results are interesting for JOF readers. This study point to the capacity of yeast form to accumulate copper as a potential application for bioremediation.
The following issues must be addressed.
Major comments
In the present study, Yarrowia lipolytica cells were cultivated in rich medium supplemented with different salts. In these conditions, it is impossible to run specification/chelator program and precisely determine free metal ion concentration in the growth medium, therefore the authors should avoid using “2 mM metal ions” (line 83 and the rest of the manuscript) and specify “2 mM CuSO4”. The authors should clearly indicate what salts were used (chloride or sulphate or others).
There are no details about how Y. lipolytica cells were grown. Usually, Yarrowia cells exhibit rigorous growth with agitation and aeration. The author should specify if they used Erlenmeyer flasks at 250 rpm or microplates. This is relevant since Y. lipolytica is strictly aerobic and oxygen supply affects fungal metabolism and influence metal/copper resistance.
It is not clear what Y. lipolytica strain was used to obtain results in Fig. 1-2. Was it W29 or TLL-expressing strain? How do you explain that control culture didn’t undergo dimorphic transition?
Yarrowia lipolytica has 16 paralog genes coding for lipases with some lipases well characterized (Kamoun et a;l, 2015 Biochim Biophys Acta. 1851(2):129-40. doi: 10.1016/j.bbalip.2014.10.012.). The authors use the protocol employed for determination of lipase from bacteria Pseudomonas – does this protocol ensure determination of fungal lipase? I wonder if other protocol would render better lipase activity. Did authors determine the endogenous lipase activity to examine its modulation upon copper stress?
Figure 3C - quantification and statistical analysis should be performed comparing TLL lipase activity.
RNA seq assay was performed using W29 wt cells grown with 1 mM CuSO4, which has only slight effect on growth/transition. It is not clear to me why this experiment was not performed with 2 mM, used for other assays.
Authors concluded (line 18) that “hyphal cells survived better than yeast-form cells with copper ions.” Taken together with the result that “the intracellular Cu(II) accumulation was drastically reduced upon hyphae formation” could it be the indication that hyphae cells exhibit strong copper detoxification by means of copper efflux using copper transporters in plasma membrane? This possibility should be addressed and discussed, also considering RNAseq data.
The authors concluded that copper “function as a signaling molecule in the dimorphic transition. To date, how Cu(ii) transmits signals remains elusive.”
Because of its transition nature, copper effect mainly explained by the increase in reactive oxygen species production and subsequent ROS effect. Copper-mediated oxidative stress, and ROS function as signaling factor in eucaryotic cells, are well documented in literature. Unfortunately, copper link to oxidative stress is disregarded in the manuscript and not used to explore the results with differentially expressed genes.
How do identified in this manuscript genes, and specifically those which expression induced dimorphic transition, relate to MAPK, PKA and Rim signaling pathway which are known to regulate dimorphic transition? This is crucial to understand the molecular mechanism(s) underlying copper-dependent dimorphic transition. The authors should propose some model and compare their data with other studies (for instance see Pomraning et al mSphere. 2018 5;3(6):e00541-18; Morales-Vargas et al. Res Microbiol 2012 163(5):378-87; Morin et al. J Mass Spectrom. 2007 42(11):1453-62)
Minor comments:
- Authors should avoid generalization on abstract and mention exactly what kind of “yeast cell activities were decreased by Cu(II) treatment and then increased during the yeast-to-hyphae transition” (lines 16-17).
- Fig. 1 - author should provide quantification of the “significantly promoted” dimorphic transition.
- Line 9 – author should not mistake CTR1 transporter and CRF1 transcription factor.
- I wonder why the authors cultivated Y. lipolytica cells at so high glucose concentration (5% instead of 2%). Does this influence copper toxicity?
- The growth medium used in this study does NOT correspond to classical YPD broth (Yeast extract 1% ; Peptone 2% ; glucose 2%). This should be corrected and the medium renamed.
- Fig 1 - should use “Control” instead of “no ion”, since the growth medium has plenty of ions.
- Fig. 5 – please verify the pathway “chemokine signaling pathway”, yeast cells do not exhibit chemokine system.
- Provide citation for KOG database.